# Myocardial Injury and Overload among Amateur Marathoners as Indicated by Changes in Concentrations of Cardiovascular Biomarkers

**DOI:** 10.3390/ijerph17176191

**Published:** 2020-08-26

**Authors:** Anna Maria Kaleta-Duss, Zuzanna Lewicka-Potocka, Alicja Dąbrowska-Kugacka, Grzegorz Raczak, Ewa Lewicka

**Affiliations:** 1Department of Cardiology and Electrotherapy, Medical University of Gdańsk, 80-210 Gdańsk, Poland; ania.m.kaleta@gumed.edu.pl (A.M.K.-D.); zuzanna.lewicka@gumed.edu.pl (Z.L.-P.); alidab@gumed.edu.pl (A.D.-K.); gracz@gumed.edu.pl (G.R.); 2First Department of Cardiology, Medical University of Gdańsk, 80-210 Gdańsk, Poland

**Keywords:** marathon, sports cardiology, athlete’s heart, biomarkers

## Abstract

Marathons continue to grow in popularity among amateurs. However, the impact of intensive exercise on the amateur’s cardiovascular system has not yet been studied. Analysis of the influence of the marathon on kinetics of biomarkers reflecting cardiac injury and overload may bring new insights into this issue. We investigated the effect of running a marathon on the concentrations of high sensitivity cardiac troponin I (hs-cTnI), heart-type fatty acid binding protein (H-FABP), N-terminal proatrial natriuretic peptide (NT-proANP), B-type natriuretic peptide (BNP), growth differentiation factor 15 (GDF-15) and galectin 3 (Gal-3) in the population of male amateur runners. The study included 35 amateur marathoners and followed 3 stages: S1—two weeks prior to the marathon, S2—at the finish line and S3—two weeks after. Blood samples were collected at each stage and analyzed for biomarkers and laboratory parameters. Concentrations of all studied biomarkers were significantly higher at S2, whereas at S3 did not differ significantly compared to S1. Running a marathon by an amateur causes an acute rise in biomarkers of cardiac injury and stress. Whether repetitive bouts of intensive exercise elicit long-term adverse cardiovascular effects in amateur marathoners needs further research.

## 1. Introduction

Extreme sports are constantly growing in popularity. Among them marathon running is the most popular, with the number of participants estimated at 2 million in 2015 [1] and over 3800 races organized worldwide in 2017. Most of the participants are amateur runners—they do not play sports for a living and present with a broad spectrum of lifestyle and cardiovascular risk factors. Even if they change their lifestyle to a “healthy” one deciding to participate in the marathon, the preparations require daily regimen and regular training with the intensity that exceeds the routines of amateur sports.

General beneficial influence of regular physical activity on cardiovascular system has been proved in numerous prospective studies that have assessed both all-cause and cardiovascular mortality in relation to levels of physical activity. Bouvier at al. examined a group of elderly veteran endurance athletes and compared their left ventricular function and myocardial perfusion with healthy moderately physically active subjects. The authors demonstrated better systolic and diastolic left ventricular function in the group of endurance athletes [2]. Randers et al. compared the cardiovascular health and exercise performance among elderly soccer players, endurance athletes, strength-trained athletes and sedentary subjects. The authors concluded that exercise capacity and cardiovascular health were the best in the group of soccer players and endurance athletes [3]. 

Furthermore, a recent meta-analysis by Lemez and Baker showed that participation in elite sport is generally favorable to lifespan longevity [4]. The above studies conclude that long-term endurance physical activity does not adversely affect the cardiovascular system, on the contrary, by increasing the systolic and diastolic function of the left ventricle and reducing the cardiovascular risk factors, it contributes to the extension of life. 

However, a question arises whether bouts of intense exercise, such as a marathon run, elicit only positive effects on the heart of an amateur runner, or whether they may cause adverse effects as well.

Numerous studies performed on elite athletes have revealed that strenuous exercise results in a physiological adaptation with symmetrical cardiac chambers’ enlargement, and reversible diastolic dysfunction was considered a cardiac fatigue phenomenon [5]. In a recently published study, Abdullah et al. showed that long-term sports activity, regardless of dose, was not associated with focal myocardial fibrosis. Therefore, the authors share the opinion that long-term endurance activity does not have a detrimental effect on the cardiovascular system [6]. However, other authors reported transient impairment of left [7] and right ventricular function [8] as well as the function of left atrium [9]. With functional abnormalities shown by visual imaging techniques come abnormal concentrations of cardiac biomarkers. It has been shown that endurance exercise evokes transient elevated concentrations of creatine kinase (CK), creatine kinase MB isoenzyme (CK-MB), cardiac troponin (cTn) and B-type natriuretic peptide (BNP) [10]. Recently, a number of novel biomarkers of cardiac injury and overload have been reported. Their kinetics have not yet been studied in the cohort of amateur athletes although they may bring new insights into understanding the influence of endurance sport on the heart in this heterogenous population. 

## 2. Materials and Methods

We examined a group of male amateur marathon runners by electrocardiography, echocardiography and blood testing. The aim of the study was to define changes in concentrations of biomarkers of cardiac injury and overload after a marathon run including novel biomarkers: N-terminal pro-atrial natriuretic peptide (NT-proANP), growth differentiation factor 15 (GDF-15) and galectin 3 (Gal-3).

The study was performed among 35 Caucasian male amateur marathoners, who finished the 2nd PZU Marathon in Gdańsk, Poland on the 15th of May 2016 on the distance of 42 km 195 m. The study protocol was approved by the Bioethics Committee of the Medical University of Gdansk (No. NKBBN 104/2016). Participation was voluntary and a written proof of consent was obtained from each participant prior to their enrollment. Only those with no history of diagnosed chronic illness were enrolled. Information on personal health and intensity of training was obtained via anamnesis. During the marathon run, the subjects were allowed to consume fluids and food ad libitum. Before the completion of the third stage of the study, the participants were asked to perform normal activities, but without strenuous exercise, including participation in any upcoming competitions.

The study was divided into three stages. The first stage (S1) was carried out 2 weeks before the run, the second stage (S2) on the finish line just after the run had been finished and the third stage (S3) 2 weeks after the marathon. First and third stages were carried out in the Department of Cardiology and Electrotherapy Medical University of Gdańsk. At each stage blood samples from cubital veins were collected. At S1 and S3, fasting blood samples were gathered. None of the samples showed any signs of hemolysis. Serum was prepared immediately after the collection by centrifugation at 2000 rpm at room temperature for 12 min and then stored in −80 °C before the final analysis.

At every stage, the samples were analyzed for hemoglobin (Hb), hematocrit (Hct), red blood cells count (RBC), creatine kinase (CK), creatinine and electrolytes—sodium (Na) and potassium (K). Concentrations of cardiac injury markers: high sensitivity cardiac troponin I (hs-cTnI) and heart-type fatty acid binding protein (H-FABP) were analyzed, along with cardiac overload biomarkers: BNP, NT-proANP, GDF-15 and Gal-3. The level of hs-cTnI was measured using Architec I2000 (Abbott) analyzer, with sensitivity of 0.0013 ng/mL, detection range to 50.00 ng/mL and lower limit for myocardial ischemia equal to 0.0342 ng/mL (for males). H-FABP level was measured via Human FABP3 DuoSet ELISA (R&D Systems), with the detection range from 0.94 to 30 ng/mL. The cut-off value for acute coronary syndrome in patients with chest pain ranged from 6.32 ng/mL to 17.7 ng/mL according to various studies [5,6]. BNP level was measured using a solid-state sandwich ELISA test (Wuhan EIAab Science), with a sensitivity of 3.9 pg/mL and detection range from 15.6 to 1000 pg/mL. The cut-off values for acute and chronic heart failure were 100 pg/mL and 35 pg/mL, respectively. NT-proANP level was measured using a solid sandwich ELISA test (R&D Systems), with detection range from 15.6 to 1000 pg/mL. Gal-3 level was measured by a solid state sandwich ELISA (R&D Systems), with a sensitivity of 0.085 ng/mL and detection range from 0.3 to 10 ng/mL. GDF-15 was measured via Human growth/differentiation factor 15 ELISA Kit (Wuhan EIAab Science, Wuhan, China) with a sensitivity of 7.8 pg/mL and detection range from 15.6 to 1000 pg/mL.

All continuous variables were expressed as mean values (standard deviation, SD). The normality of each variable was tested by the Shapiro–Wilk test. Statistical differences between groups of variables were tested using the analysis of variance (ANOVA) for repeated measures. Tukey’s honest significant difference test was used in the post-hoc analysis. Correlations were measured using the Pearson correlation coefficient. The collected data was analyzed using the Statistica 12 software (StatSoft, Kraków, Poland). A *p*-value < 0.05 was found to be statistically significant.

## 3. Results

### 3.1. Study Group

We enrolled 40 Caucasian male amateur marathon runners aged between 22 and 55 years, 39 (8) years. Among them, 35 runners finished the marathon run and their blood samples were obtained at each stage for analysis. During the marathon run, the average ambient temperature and humidity were 16–18 degrees Celsius and 50–60%, respectively. Demographic data of participants, the intensity of training before the run and the marathon completion time are shown in Table 1.

### 3.2. Biochemical Analysis

The results of biochemical analysis are presented in Table 2. 

The mean hemoglobin (Hgb) concentration at S1 was 15.1 g/dL. There was no significant difference in Hgb between S1 and S2. The lowest mean Hgb of 14.5 g/dL was found at S3, which differed significantly from both S1 and S2. Hematocrit (Hct) differed significantly between all the stages, with the highest value measured at S2 and the lowest at S3. The mean concentration of CK was significantly higher at S2: 411 (170) U/L compared to S1 and S3, and for 19 participants it exceeded the upper limit of normal range (200 U/L). There were no significant differences in CK level between S1 and S3. The mean creatinine concentration was at its peak immediately after the marathon: 0.89 (0.10) mg/dL and it was significantly higher compared to S1, whereas it did not differ significantly from S3. There were also no significant differences in creatinine concentration between S1 and S3. The highest plasma sodium level (Na): 142 (2) mmol/L, was found at S2, and it was significantly higher from both S1 and S3. The mean Na concentration did not differ between S1 and S3. Neither hyper- nor hyponatremia were observed among the participants. The mean plasma potassium (K) concentration was at its lowest at S3: 4.1(0.2) mmol/L; however, the differences between the stages were not significant.

### 3.3. Cardiac Biomarkers

The results of the biomarkers’ concentration analysis are shown in Table 3. 

Mean hs-cTnI concentration at S2 was 0.06 ng/mL, which was significantly higher compared to S1, and 30 participants presented values higher than at S1. In 15 participants (42.8%) hs-cTnI levels were above the cut-off for myocardial ischemia (>0.0342 ng/mL). The differences between S1 and S3 were not significant.

Mean BNP concentrations were at their highest at S2 and they differed significantly from S1 and S3. In 21 participants the BNP level at S2 was higher compared to S1. In 16 participants (45.7%), BNP >100 pg/mL was found at S2. 

Mean NT-proANP, Gal-3 and GDF-15 concentrations were at their highest at S2. They were significantly higher compared to both S1 and S3, whereas they did not differ significantly between S1 and S3. At S2, in 26 participants the NT-proANP concentrations were higher compared to the baseline, Gal-3 in 23 participants and GDF-15 in 25 participants, respectively.

### 3.4. Post-Race Change in the Analyzed Parameters

When compared to S1, we found various degrees of changes in the concentrations of investigated parameters (Table 4). 

On the finish line, the mean concentrations of all measured parameters were increased. The most remarkable rise was found for H-FABP and hs-cTnI concentrations, of 511 and 500%, respectively. The most stable parameters were electrolytes, Hgb, RBC count and Hct. At S3, the greatest decrease was found for concentrations of hs-cTnI (−57%) and H-FABP (−34%). Decrease in concentration of NT-proANP was by 7.28%. Interestingly, BNP, GDF-15 and Gal-3 remained elevated compared to S1; however, the differences were not statistically significant. Despite percentage differences, concentrations of all biomarkers at S3 did not differ significantly compared to S1 (see: Table 3).

We found a strong positive linear correlation between the concentrations of hs-cTnI and H-FABP (r = 0.55), as well as Gal-3 and H-FABP (r = 0.52) (Table 5). 

Interestingly, a negative linear correlation between hs-cTnI concentration and age was observed. There were no significant correlations between the concentrations of biomarkers at the finish line (S2) and the intensity of training (km run per week or hours run per week) or the time of the marathon run. 

## 4. Discussion

Our study has been the first one to analyze changes in novel biomarkers reflecting cardiac injury and overload in the group of male amateur runners and assessed at baseline, immediately post-marathon and two weeks after the marathon. We found that completing a marathon by an amateur led to an acute, significant cardiac volume and pressure overload, as indicated by significant increases in BNP, NT-proANP and GDF-15 levels. Furthermore, a significant rise of hs-cTnI and H-FABP concentration could be suggestive of transient myocardial ischemia. In addition, the positive correlation found between H-FABP and Gal-3 may indicate the relationship between exercise-induced ischemia and cardiac remodeling. 

The question about the dose of physical activity, which is beneficial for the cardiovascular system, seems even more relevant in times of the growing popularity of extreme sports among amateurs. The amount of exercise, recommended by the European Guidelines on Cardiovascular Disease Prevention in Clinical Practice, is 150 min a week of moderate intensity, 75 min a week of intensive aerobic physical activity or an equivalent combination of both [11]. 

In our study group, the average number of running hours per week was 6.2, which significantly exceeds values recommended in the guidelines. The mean time of marathon completion among our participants was below 4 h (234 min). Given that our participants have never trained sport at a professional level, this result shows that the boundaries of “amateur sport” are nowadays set much higher compared to the past and that people train “aggressively” to achieve more ambitious goals. Sharma et al. [12] analyzed acute events in endurance athletes (cardiac arrests, exercise-related collapse, heat stroke, chest pain, abdominal pain and suspected fractures) based on 10 years of experience in the London Marathon and showed that in a group of middle-aged male marathon runners with a three to four hour completion time, the risk was the highest. 

Lee et al. [13], who studied the dose of exercise in the context of cardiovascular disease prevention, showed that people who ran longer (>2.5 h/week) or at a higher speed had a risk of mortality increased from three to nine times compared to light joggers.

In our study of amateur marathon runners, we found a significant, transient increase in biomarkers for myocardial injury: hs-cTnI and H-FABP immediately after the marathon. What is more, their increase was the most significant of all the analyzed parameters: 500% for hs-cTnI and 511% for H-FABP. The mean cTnI level at S2 was 0.06 ng/mL and 15 participants (42.8%) presented with hs-cTnI levels above the cut-off for myocardial ischemia (>0.0342 ng/mL) at S2. H-FABP is an ultra sensitive marker of myocardial damage and its increase corresponds with the increase in cardiac troponins, which was also found in this study. The correlation between hs-cTnI and H-FABP was the strongest of all measured parameters, with r = 0.55 (see: Table 5). 

The question arises, whether such an acute elevation of cardiac troponins and H-FABP in an amateur can be considered as a physiological response of myocardium to intense exercise or if it is pathological. 

Our doubts are reflected in other studies. Sherr et al. [14] who studied the kinetics of cardiac biomarkers, including troponin and H-FABP, showed that although immediately after the race the concentrations of cTnI and H-FABP were significantly elevated, they returned to normal ranges within 72 h. The authors associated these changes rather with an altered cardiomyocyte metabolism than cardiac damage. In another study, Scharhag et al. [10] indicated that the increase in cardiac troponin, if mild and transient, probably reflects the reversible cardiomyocyte membrane leakage due to heavy effort with the release of troponin from the free cytosolic pool, and thus their elevated plasma concentration should be considered physiological.

It has to be mentioned that both studies included the population of elite athletes and, what is more, the authors emphasize the need for further research in the field.

Legaz-Arrese et al. [15] investigated cTnI in elite and amateur athletes after 30 min of intensive rowing. They found that although the elite group showed higher baseline and post-exercise levels of cTnI (0.019 ng/mL and 0.080 ng/mL, respectively) compared to amateurs (0.015 ng/mL and 0.030 ng/mL), the post-exercise increase in both groups was similar (around 4-fold). This observation shows that the trends observed in athletes may also apply to amateurs. 

Heidbuchel et al. [16] hypothesized that in a small percentage of athletes, excessive exercise combined with insufficient regeneration periods can lead to accumulation of microdamages and, as a consequence, to so-called exercise-induced cardiomyopathy. Keeping in mind the fact that amateur athletes generally do not follow professional training plans, they may not strictly adhere to the regeneration intervals. As a consequence, amateurs may be particularly susceptible to microdamages of myocardium. Moreover, the link between the incidence of atrial fibrillation and endurance sports has widely been studied using visualization techniques. Elliot et al. in their recent cohort study on the group of 99 recreational endurance athletes who were grouped according to lifetime training hours, found that left atrial volume was significantly greater in the group of high and medium lifetime training hours compared to the low group. There were no differences in left ventricular dimensions or function or the incidence of premature atrial or ventricular beats. The authors concluded that increased lifetime training is associated with left atrial dilatation which may promote the occurrence of atrial fibrillation in this cohort [17,18]. Nevertheless, due to the short observation period in our study, the hypothesis of exercise-induced cardiomyopathy and the predisposition to develop atrial fibrillation in amateur runners cannot be tested. 

BNP is a cardiac neurohormone released from the ventricles under conditions of elevated pressure and stress, which reduces cardiac load by natriuresis and vasodilatation. In conditions of increased cardiac volume and pressure, like a marathon run, BNP levels rise. Moreover, BNP is believed to elicit antiproliferative effects on myocardium, as indicated in experimental studies where BNP gene knockout leads to myocyte proliferation and fibrosis. In humans, however, this has not yet been proved [19]. 

In our study group, the mean BNP at S2 was 155.38 pg/mL and was significantly higher compared to both S1 and S3, with the maximum of 574.4 pg/mL. BNP is widely used to diagnose and monitor acute and chronic heart failure with a cut-off limit of 100 pg/mL in acute heart failure. In our study group, 16 participants (45.7%) presented BNP > 100 pg/mL at S2. Similarly, Roca et al. [20] also found increased NT-proBNP levels and above a limit for acute heart failure in 30.7% of nonelite athletes immediately post-race. Klenk et al. [21] studied the clinical significance of the exercise-induced release of cTn and NT-proBNP in the population of endurance athletes. In spite of the acute elevation of biomarkers’ concentrations, no functional damages of myocardium investigated by visualizing techniques (echocardiography and magnetic resonance) were found. In another study, the authors suggested that the increased concentration of BNP after an endurance exercise may have cytoprotective and growth-regulating effects and should not be regarded as pathological [22]. 

Pro-atrial natriuretic peptide (proANP) is a marker of an increase in atrial wall tension, and its higher concentrations have been reported in subjects with atrial fibrillation [23]. Wilhelm et al. [24] found a strong positive correlation between proANP concentration at baseline and after a 10-mile race and right atrial volume among marathon runners compared to nonmarathon runners, which is consistent with electrocardiographic findings from our previous study [25] and emphasizes the importance of right atrial overload following a bout of strenuous exercise. 

Our study is the first assessing N-terminal proatrial natriuretic peptide (NT-proANP) concentrations in a population of amateur marathon runners. NT-proANP is a biologically inactive fragment (98 amino acids) of ANP prohormone and has been reported as a useful diagnostic and prognostic tool for patients with heart failure and myocardial infarction. In a recent experimental study, it was shown that NT-proANP was excellent for indicating cardiac hypertrophy [26]. 

In our study, at S2 the mean NT-proANP concentration was 60.54% higher compared to the baseline and it corresponded with the increase in BNP level (79.74%) indicating that during intensive exercise both ventricles and atria remain under significant volume and pressure overload. Future studies will determine whether NT-proANP may indicate a tendency to atrial fibrillation in subjects who practice sports intensively. 

Until now, GDF-15 has not yet been studied in the context of amateur marathon running. Its synthesis is triggered by cardiac ischemia, inflammation or injury, and elevated concentrations indicate a poor prognosis in patients with acute coronary syndrome and heart failure [27]. Lankeit et al. [28] showed that GDF-15 level at admission was an independent predictor of long-term mortality in patients with acute pulmonary embolism, which is a state of extreme pressure overload for the right atrium and right ventricle. Galliera et al. [29] reported a significant increase in GDF-15 levels due to intense rugby training, which is consistent with our findings. We found a significant increase in GDF-15 level at S2 of 169.45% compared to baseline, which may indicate acute cardiac hemodynamic volume and pressure overload immediately after the marathon.

Galectin 3 (Gal-3) was reported as a marker of ventricular remodeling through myofibroblast proliferation in response to an increased left ventricular tension. Increase in its concentration was observed in patients with acute and chronic heart failure [30]. Our study is the first that reports Gal-3 kinetics in the context of intensive exercise in amateurs. It should be noted that a correlation between H-FABP and Gal-3 was found; however, the potential significance of this finding cannot yet be determined and requires further research. 

On the finish line, the limit for the diagnosis of exertional muscle damage in some participants was exceeded three-fold, with the highest creatine kinase level of 858 U/l, whereas others presented with CK concentrations within normal range. This may be due to polymorphisms of genes regulating the inflammatory response, like IL-6 and mainly TNF-alpha, which is associated with suppression of protein synthesis in skeletal muscle and enhancement of catabolic pathways [31]. Interestingly, in the studied group, two participants with the highest CK concentrations at S2 also presented with the highest concentrations of cardiac troponin I. This observation may suggest a common pathophysiological mechanism of exertional damage to skeletal muscles and heart in some individuals whose genetic polymorphisms induce excessive inflammatory reactions in response to intense physical activity.

In our study, it should be noted that the concentrations of all investigated biomarkers returned to levels compared to the baseline 14 days after the marathon. In addition, changes in the concentrations of new biomarkers and their normalization were similar to those previously reported for troponin or natriuretic peptides [9]. This observation corresponds to the results of the other studies where the authors conclude that the effect of the intensive endurance exercise is only transient and does not indicate cardiac damage but reflects a physiological response to a state of acute stress and volume overload. 

In a recent review by Stavroulakis and George [32], exercised-induced release of cardiac troponin was considered a benign phenomenon, mainly because there was no evidence of a permanent myocardial dysfunction. Nevertheless, it has to be mentioned that the authors analyzed a remarkably diverse population of athletes practicing a broad spectrum of activities from walking to ultraendurance sports. Undoubtedly, exercise brings many benefits and prolongs life. Regular physical activity can counteract the negative effects of lifestyle and other cardiovascular risk factors. The EXCITE study [33] showed a beneficial effect of exercise on the development of coronary collateral circulation, which may have some preconditioning effect not only in subjects diagnosed with coronary artery disease.

Finally, our study has several limitations that should be mentioned. Firstly, it was conducted on a relatively small group including male subjects only. Secondly, it was a cross-sectional study and the observation time was limited to two weeks after the run. We did not monitor the kinetics of biomarkers in the time interval between S2 and S3 as well as among the female population. Thirdly, we did not include echocardiographic imaging, but this was not the purpose of this part of the study. 

## 5. Conclusions

The population of amateur athletes has so far been poorly represented in previous studies on intense endurance exercise and its effects on the heart. The present study is the first to investigate the impact of the marathon run on the concentrations and kinetics of novel cardiac injury and stress biomarkers, including NT-proANP, GDF-15 and Gal-3, in the male amateur marathon population. Completing a marathon by an amateur leads to an acute, significant cardiac volume and pressure overload, confirmed by a significant increase in the concentrations of BNP, NT-proANP and GDF-15. Significant elevations of hs-cTnI and H-FABP show that an intensive bout of exercise leads to a transient myocardial oxygen supply-demand mismatch. Moreover, the positive correlation between H-FABP and Gal-3 may indicate the link between exercise-induced cell injury and cardiac remodeling. 

However, the concentrations of all investigated biomarkers returned to levels compared with the baseline 14 days after the marathon. This observation is concurrent with the results of the previous studies on elite athletes and may suggest that the acute alterations of biomarkers mirror the physiological myocardial response to the acute pressure and volume overload in terms of cardiac fatigue. However, all these studies were cross-sectional and carried out on small groups of highly selected subjects. That is why we believe that although there are many premises to conclude that intensive endurance exercise results only in cardiac fatigue, until the results of longitudinal studies with long follow up periods are available, it is too early to state that there are no long-term adverse effects. Some authors suggest that the population of endurance athletes is heterogenous in itself, so that chronic bouts of intense exercise in one athlete lead to marked but healthy adaptation, whereas in another—to heart injury. This also applies to amateur athletes.

We have found that individuals who are generally considered amateurs represent the same trend of the concentrations of cardiovascular biomarkers as elite athletes. On the other hand, amateurs constitute a widely heterogeneous group in terms of general lifestyle, cardiovascular risk factors and individual, not professionally designed training regimens. Therefore, they may be more prone to developing adverse effects of a long-term high-intensity stimulus on the heart compared to elite athletes. Potential predisposing factors need to be further investigated. 

Currently there is insufficient evidence to conclude that intensive endurance exercise elicits long term adverse effects on the heart of an amateur runner. It has to be emphasized that the general significance of physical activity to public health should not be diminished, as the potential negative effects of the heart—if clearly proved—would be clinically relevant only to a few. Nevertheless, we see the need for a cardiological minimum in individuals who are planning to practice intensive endurance sports: a medical check-up including medical history and cardiovascular risk profile and resting ECG.

## Figures and Tables

**Table 1 ijerph-17-06191-t001:** Demographics, training intensity and results of the marathon run in the studied group.

Characteristics		Marathon Runners (N = 35)
Demographics		
	Age [years]	39 ± 8 ^1^
	Gender	35 males
	BMI [kg/m^2^]	25 ± 2
	Ethnicity	35 Caucasian
	Smokers/nonsmokers	35 nonsmokers
Training intensity		
	hours of running/week	6.2 ± 2.3
	kilometers run/week	54.5 ± 18.6
Result of the marathon		
	Duration [min]	234 ± 25

^1^ All continuous data presented as mean ± SD; BMI: Body Mass Index

**Table 2 ijerph-17-06191-t002:** Results of biochemical analysis performed 2–3 weeks before the marathon (S1), on the finish line (S2) and 2–3 weeks after the marathon (S3).

Parameter	Laboratory Norms	S1	S2	S3	ANOVA	Post-Hoc *p*-Value		
			Mean ± SD		*p*-Value	S1 vs. S2	S2 vs. S3	S1 vs. S3
Hemoglobin [g/dL]	13.0–17.0	15.1 ± 1.0	15.3 ± 0.9	14.5 ± 0.9	<0.001	0.17	<0.001	<0.001
Hematocrit [%]	40–50	43.5 ± 2.4	44.4 ± 2.3	41.6 ± 2.6	<0.001	0.02	<0.001	<0.001
Red blood cells [T/L]	4.5–5.5	5.1 ± 0.3	5.2 ± 0.3	4.9 ± 0.3	<0.001	0.01	<0.001	<0.001
Creatine kinase [U/l]	30–200	148 ± 76.3	411 ± 170	208 ± 135	<0.001	<0.001	<0.001	0.09
Creatinine [mg/dL]	0.6–1.3	0.75 ± 0.09	0.89 ± 0.1	0.80 ± 0.06	0.02	0.02	0.08	0.81
Sodium [mmol/L]	136–145	140 ± 2	142 ± 2	139 ± 2	<0.001	<0.001	<0.001	0.36
Potassium [mmol/L]	3.5–5.1	4.2 ± 0.3	4.2 ± 0.5	4.1 ± 0.2	0.14	-	-	-

ANOVA: analysis of variance.

**Table 3 ijerph-17-06191-t003:** Concentrations of cardiac injury and overload biomarkers: 2–3 weeks before the marathon (S1), on the finish line (S2) and 2–3 weeks after the marathon (S3).

Parameter	S1	S2	S3	ANOVA	Post-Hoc *p*-Value		
		Mean ± SD		*p*-Value	S1 vs. S2	S2 vs. S3	S1 vs. S3
hs-cTnI [ng/mL]	0.01 ± 0.01	0.06 ± 0.09	0.00	<0.001	<0.001	<0.001	0.98
H-FABP [ng/mL]	2.22 ± 1.18	13.57 ± 9.63	1.47 ± 0.84	<0.001	<0.001	<0.001	1.00
BNP [pg/mL]	79.86 ± 53.11	155.38 ± 156.23	84.69 ± 50.01	<0.001	0.001	0.004	0.94
NT-proANP [pg/mL]	469.25 ± 155.44	753.3 ± 176.60	435.09 ± 170.79	<0.001	<0.001	<0.001	0.31
Gal-3 [ng/mL]	8.53 ± 3.04	10.65 ± 2.33	8.78 ± 1.67	<0.001	0.001	0.004	0.92
GDF-15 [pg/mL]	50.97 ± 27.61	137.34 ± 85.19	51.67 ± 24.45	<0.001	<0.001	<0.001	0.97

hs-cTnI: high sensitivity cardiac troponin I, H-FABP: heart-type fatty acid binding protein, BNP: B-type natriuretic peptide, NT-proANP: N-terminal proatrial natriuretic peptide, Gal-3: galectin 3, GDF-15: growth differentiation factor 15.

**Table 4 ijerph-17-06191-t004:** Post-race changes in concentrations of hematological and biochemical parameters and biomarkers expressed as a percentage change from baseline (S1).

Parameter	S2	S3
	[%]	
Hemoglobin	1.32	−3.97
Hematocrit	2.07	−4.37
Red blood cells	1.96	−3.92
Creatine kinase	177.7	40.54
Creatinine	18.7	6.67
Sodium	1.43	−0.71
Potassium	1.41	−2.54
hs-cTnI	500	−56.76
H-FABP	511.26	−33.78
BNP	79.74	20.37
NT-proANP	60.54	−7.28
Gal-3	24.85	2.93
GDF-15	169.45	1.37

**Table 5 ijerph-17-06191-t005:** Correlations between biomarkers, age and time of run at S2.

	hs-cTnI	H-FABP	BNP	NT-proANP	Gal-3	GDF-15	Time of Run
Age	−0.43 ^a^	−0.28	0.27	−0.16	−0.22	0.26	0.03
Time of run	0.21	0.18	0.10	−0.25	−0.14	−0.37	
GDF-15	−0.05	0.17	−0.22	0.02	0.12		
Gal-3	0.18	0.52 ^b^	−0.35	−0.03			
NT-proANP	−0.04	−0.02	0.27				
BNP	−0.23	−0.10					
H-FABP	0.55 ^c^						

^a^*p* = 0.01 ^b^
*p* = 0.003 ^c^
*p* = 0.001.

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
