# Peer review of "Myocardial Injury and Overload among Amateur Marathoners as Indicated by Changes in Concentrations of Cardiovascular Biomarkers"

_ijerph, 2020, doi:10.3390/ijerph17176191_

Round 1

Reviewer 1 Report

The manuscript is well written and your study is interesting. It provides interesting information about
molecular biomarkers associated with physical exercise in amateur athletes. Often has been published
studies about the echocardiographic modification induced by physical exercise and for this reason your
article is innovative. However I would suggest you some minor revision:
1) Re-write the references’ number order because some numbers are missing.
2) Define the abbreviations the first time they appear in the text, in the abstract and in tables and
figures.
3) You often underline the bad effect of physical activity on myocardium. In your opinion may the
stress induced by exercise on myocardium induce a sort of preconditioning, thus, protecting heart
from possible ischemic damage induced by coronary artery disease or microvascular dysfunction
when an if they appear in future? considering the beneficial effects of physical exercise on
cardiovascular risk factors and coronary resistances regulation, may its related benefits overcome
the negative effects that you described? You should discuss about these two important points (see
Int J Mol Sci 2020 Apr 30;21(9):3167. doi: 10.3390/ijms21093167----J Diabetes Res 2019 Apr
4;2019:9489826.doi: 10.1155/2019/9489826)
4) “patomechanisms”: it is better pathophysiological mechanisms

Author Response

We would like to express our thanks for a thorough and detailed revision of our manuscript.

Please, find answers to Reviewers below:

1) Re-write the references’ number order because some numbers are missing.

The references` number order was rewritten and corrected.

2) Define the abbreviations the first time they appear in the text, in the abstract and in tables and figures.

The abbreviations in the text, in the abstract and in tables were corrected.

3) You often underline the bad effect of physical activity on myocardium. In your opinion may the stress induced by exercise on myocardium induce a sort of preconditioning, thus, protecting heart from possible ischemic damage induced by coronary artery disease or microvascular dysfunction when an if they appear in future? Considering the beneficial effects of physical exercise on cardiovascular risk factors and coronary resistances regulation, may its related benefits overcome the negative effects that you described? You should discuss about these two important points (see Int J Mol Sci 2020 Apr 30;21(9):3167. doi: 10.3390/ijms21093167----J Diabetes Res 2019 Apr 4;2019:9489826.doi: 10.1155/2019/9489826)

Undoubtedly, exercise brings many benefits and prolongs life. Regular physical activity can counteract the negative effects of lifestyle and other cardiovascular risk factors, it reduces the body mass, blood glucose concentration and blood pressure. Physical activity increases the contractility of myocardium by reducing the duration of action potential of cardiomyocytes, and therefore plays a protective role  against ischemia [1].  Wang at al. [2] in the experiments performed on rats showed that exercise training upregulates the ATP-sensitive K+ channels across the left ventricle in both sexes. Increased contractility and reduced APD helps protect cardiac output and reduce intracellular Ca2+ overload during stresses such as regional ischemia.

The EXCITE study [3] showed a beneficial effect of exercise on the development of coronary collateral circulation in subjects with stable coronary artery disease. The observations in the study were made after a short 4 week follow up.

We agree that regular physical activity may have some preconditioning effect, not only in subjects diagnosed with coronary artery disease. On the other hand, it has been proved that intensive bouts of exercise may result in the disruptions of the cytoarchitecture of myocardium (fibrosis) and in turn facilitate ventricular and supraventricular arrhythmias. This pathophysiological mechanism is believed to link long lasting endurance sport activity and incidence of atrial fibrillation.

4) “patomechanisms”: it is better pathophysiological mechanisms

The suggested expression was used.

References:

  1. Severino P et al. Ischemic Heart Disease and Heart Failure: Role of Coronary Ion Channels. I J. Mol. Sci. 2020, 21(9), 3167.
  2. Wang X, Fitts RH. Effects of regular exercise on ventricular myocyte biomechanics and K ATP channel function. Am J Physiol Heart Circ Physiol 2018 Oct 1;315(4):H885-H896.
  3. Möbius-Winkler S, Uhlemann M et al. Coronary Collateral Growth Induced by Physical Exercise. Results of the Impact of Intensive Exercise Training on Coronary Collateral Circulation in Patients With Stable Coronary Artery Disease (EXCITE) Trial. 2016;133:1438–1448.

Reviewer 2 Report

The present work has very carefully examined numerous known and recent cardiac biomarkers for possible or apparent „myocardial damage“ before, shortly after and a few days after a marathon run in leisure time athletes. Results show that in almost all of these runners the markers were elevated and returned to normal values after a few days . It is a purely cross-sectional study, long-term observations were not made, so that damage to the heart cannot be proven.
Many studies on this subject have been carried out in recent years and decades on this subject, a number of critical analyses (Levine, Sanchis -Gomar and others )have shown that these findings of biomarkers do not indicate damage of the heart, but heart fatigue comparable to muscular exhaustion. No permanent damage occurred in previously healthy people. The first to describe this phenomenon of cardiac fatigue after exhaustive exercise were Saltin and Grimby ( ). Contrary to the observations in this study, all more recent analyses have shown that athletes have a longer life expectancy (4 large studies), are less likely to suffer from heart disease. Many analyses, e.g. CAC in athletes confirmed this thesis.

The authors have not or insufficiently considered most of these studies. In addition, there are publications that show unequivocally that after an exhaustive exercise , even in extreme athletes, the bio-markers quickly normalised, the fitter the runner, the faster the recovery.

In addition, long-term studies are obligatory for the detection of healthy or sick runners. The authors of this study did not do this and did not consider these factors mentioned above.

There work can therefore not be accepted for publication, at best after very thorough revision in appreciation of the publications mentioned above demonstrating no harm to the heart after exhasutive excercise in otherwise healthy athletes.

Some of the "counter-arguments" are listed in the appendix, shortly translated from a german paper.

A corresponding revision with a presentation of the "counter-arguments" is necessary if the work will be considered to be accepted.

The results thus "only" confirm that the measured changes are an expression of cardiac fatigue and have no pathological significance in otherwise healthy athletes.

This thesis is supported by numerous studies with biomarkers, heart ultrasound and also MRI.

You can refer to the attached paper.

Author Response

We would like to express our thanks for a thorough and detailed revision of our manuscript.

Please, find answers to Reviewers below:

1)The present work has very carefully examined numerous known and recent cardiac biomarkers for possible or apparent „myocardial damage“ before, shortly after and a few days after a marathon run in leisure time athletes. Results show that in almost all of these runners the markers were elevated and returned to normal values after a few days. It is a purely cross-sectional study, long-term observations were not made, so that damage to the heart cannot be proven.

We agree that one of the major limitations of our study is the lack of long-term follow up. The most frequently studied biomarker has been cardiac troponin (cTn). However, it remains only a hypothesis that an acute and temporary elevation in cTn concentration is part of the physiological response to exercise. The mechanisms of cTn-shift into the intravascular space are not known. That is why, some authors link these acute, temporary rises in cTn concentration post-exercise (and other biomarkers, like NT-pro-BNP) with the reduction in cardiac function, especially in relation to the right ventricle [1].

Many studies on this subject have been carried out in recent years and decades on this subject, a number of critical analyses (Levine, Sanchis -Gomar and others) have shown that these findings of biomarkers do not indicate damage of the heart, but heart fatigue comparable to muscular exhaustion. No permanent damage occurred in previously healthy people. The first to describe this phenomenon of cardiac fatigue after exhaustive exercise were Saltin and Grimby. Contrary to the observations in this study, all more recent analyses have shown that athletes have a longer life expectancy (4 large studies), are less likely to suffer from heart disease. Many analyses, e.g. CAC in athletes confirmed this thesis. The authors have not or insufficiently considered most of these studies.

We agree, that many authors relate to `cardiac fatigue` as a physiological phenomenon expressing the myocardial adaptation to intensive exercise. In the revised version of the manuscript, we mention this important argument. We also, as suggested, included the data about longevity of athletes.

However, other studies emphasize the effect of intensive exercise especially on the right ventricle. Here  we would like to mention the study of La Gerche et al. who found that 5 out of 40 (12.8%) experienced middle-aged endurance athletes had cardiac magnetic resonance evidence of myocardial fibrosis, and athletes with scarring demonstrated significantly larger right ventricles with lower indices of right ventricular systolic function in comparison with athletes without scarring [2].

On the other hand, there is a research of Bohm et al. who demonstrated myocardial fibrosis by cardiac magnetic resonance in only 1 out of 33 veteran elite-level endurance athletes (which was, by the way, attributed to the previous episode of perimiocarditis) [3]. In a recent meta-analysis on the subject, Wasfy and Baggish [4] emphasize the ambiguity of results and state that it is currently unclear whether intense endurance exercise has a negative long-term  on the heart.

We believe that biomarkers like H-FABP or Gal-3 can be correlated with the earliest stages of myocardial fibrosis, which in turn may be significant after repeated bouts of intensive endurance exercise.  

In addition, long-term studies are obligatory for the detection of healthy or sick runners. The authors of this study did not do this and did not consider these factors mentioned above.

We agree, that long-term observations are lacking in the group of amateur athletes and we have mentioned this fact in the “limitations of the study”. However we would like to underline that long-term observation was not the aim of our investigation. In our opinion one marathon run resulting in “cardiac fatigue” with transient biomarkers elevation cannot shorten the life expectancy of the runner. On the contrary repeated bouts of extensive endurance exercise, especially during a short period of time in amateurs, who suddenly change their lifestyle and start practicing intensive sport can have a deleterious effect on their health. Biomarker’s leakage related to the marathon run can be an early “warning” of this possible threat.

A corresponding revision with a presentation of the "counter-arguments" is necessary if the work will be considered to be accepted.

In response to this comment, we revised the manuscript and included both studies that found that the acute rise in biomarkers reflected a physiological response to exercise, and studies that found no functional changes in myocardium over extended follow-up. We have also included the results of research on the longevity of elite athletes (we would like to express our thanks for the attached Appendix). We agree that these `counter-arguments` are necessary for understanding the complexity of the problem. We highlighted that we are by no means trying to deny the overall beneficial effect of exercise on the cardiovascular system. At the same time, we would like to emphasize that almost all research on this topic was performed on groups of elite athletes, not amateurs, as in our study.

The results thus "only" confirm that the measured changes are an expression of cardiac fatigue and have no pathological significance in otherwise healthy athletes. This thesis is supported by numerous studies with biomarkers, heart ultrasound and also MRI.

Despite the hypothesis of cardiac fatigue, in the recent years there have been studies suggesting that chronic endurance exercise contributes to unfavourable remodeling of the right ventricle. Heidbuchel et al. [5] observed a high incidence of serious arrhythmic events, including SCD (20%), in 46 young athletes with frequent ventricular ectopy or non-sustained ventricular tachycardia over a 5-year follow-up. La Gerche et al.2] postulated that the exercise dose required for arrhythmogenic right ventricular remodeling is probably > 20 h/week for > 20 years. We do not deny the beneficial effect of sport, but in our opinion we still not know where are the limits of “healthy” sport and biomarkers’ leak after the marathon can be an early sign that in some amateur runners this level of exhaustion can be too high.

References:

  1. La Gerche A, Connelly KA, Mooney DJ, et al. Biochemical and functional abnormalities of left and right ventricular function after ultra-endurance exercise. Heart 2008; 94:860–6.
  1. La Gerche A, Burns AT, Mooney DJ, Inder WJ, Taylor AJ, Bogaert J, Macisaac AI, Heidbüchel H, Prior DL. Exercise-induced right ventricular dysfunction and structural remodelling in endurance athletes.Eur Heart J. 2012; 33:998–1006.
  2. Bohm P, Schneider G, Linneweber L, Rentzsch A, Kramer N, Abdul-Khaliq H, Kindermann W, Meyer T, Scharhag J. Right and left ventricular function and mass in male elite master athletes: a controlled contrast-enhanced cardiovascular magnetic resonance study.Circulation. 2016; 133:1927–1935.
  3. Wasfy M, Baggish A. Endurance Exercise and the Right Ventricle: Weak Link, Innocent Bystander, or Key Ingredient? Circulation. 2016;133:1913–1915.
  4. Heidbuchel H, Hoogsteen J, Fagard R, et al.High prevalence of right ventricular involvement in endurance athletes with ventricular arrhythmias: role of an electrophysiological study in risk stratification. Eur Heart J 2003;24:1473–80.

Round 2

Reviewer 2 Report

The paper has been improved but there remain some problems.

The authors should also consider some papers againstthe hypothesis of cardiac damage induced by endurance exercise (see papers by Bengt Saltin et al., 1987,Ben.Levine in Circulation, by Fabian Sanches -Gomar et al.) They have not explained why then many papers demonstrate increased longevity in trained athletes. This is in controversy to authors conclusion.

They also should shortly comment that only longtime studies are abel to solve the problem they discuss.

The observed blood values primarily indicate cardiac fatigue but nor cardiac damage.

Author Response

Thank you very much for re-reviewing our manuscript. Please, find the responses to the Reviewer’s comments below:

The authors should also consider some papers againstthe hypothesis of cardiac damage induced by endurance exercise (see papers by Bengt Saltin et al., 1987, Ben.Levine in Circulation, by Fabian Sanches -Gomar et al.) They have not explained why then many papers demonstrate increased longevity in trained athletes.

In the introduction we mentioned the papers by Bouvier, Saltin et al., Abdullah et al. and by Randers et al. that show no evidence of a structural or functional myocardial abnormality in the populations of veteran endurance athletes as an argument supporting the thesis of cardiac fatigue. We explained that increased longevity among veteran athletes is due to increased systolic and diastolic left ventricular function, as well as a global reduction of cardiovascular risk factors in this population.

This is in controversy to authors conclusion.

They also should shortly comment that only longtime studies are able to solve the problem they discuss.

The observed blood values primarily indicate cardiac fatigue but nor cardiac damage.

We mentioned in the discussion that numerous studies support the hypothesis of cardiac fatigue. We emphasized that there is a lack of data to conclude that intensive endurance exercise causes long-term detrimental effects on the heart, as there are no long-term observational studies that are the only way to bring new insight into this issue. Moreover, we emphasized that the overall benefits of physical activity should by no means be undermined, as the potential negative effects of the heart – if clearly proved – would be clinically relevant only to a few.